# Radiotherapy for Mobile Spine and Sacral Chordoma: A Critical Review and Practical Guide from the Spine Tumor Academy

**DOI:** 10.3390/cancers15082359

**Published:** 2023-04-18

**Authors:** Kristin J. Redmond, Stephanie K. Schaub, Sheng-fu Larry Lo, Majid Khan, Daniel Lubelski, Mark Bilsky, Yoshiya Yamada, Michael Fehlings, Emile Gogineni, Peter Vajkoczy, Florian Ringel, Bernhard Meyer, Anubhav G. Amin, Stephanie E. Combs, Simon S. Lo

**Affiliations:** 1Department of Radiation Oncology and Molecular Radiation Sciences, The Johns Hopkins University, Baltimore, MD 21287, USA; 2Department of Radiation Oncology, The University of Washington, Seattle, WA 98195, USA; 3Department of Neurosurgery, Donald and Barbara Zucker School of Medicine at Hofstra, Hempstead, NY 11549, USA; 4Department of Radiology, The Johns Hopkins University, Baltimore, MD 21287, USA; 5Department of Neurological Surgery, The Johns Hopkins University, Baltimore, MD 21287, USA; 6Department of Neurosurgery, Memorial Sloan Kettering Cancer Center, New York, NY 10065, USA; 7Department of Radiation Oncology, Memorial Sloan Kettering Cancer Center, New York, NY 10065, USA; 8Department of Neurosurgery, University of Toronto, Toronto, ON M5T 1P5, Canada; 9Department of Radiation Oncology, The Ohio State University, Columbus, OH 43210, USA; 10Department of Neurosurgery, Charite University Hospital, 10117 Berlin, Germany; 11Department of Neurosurgery, University Medical Center Mainz, 55131 Mainz, Germany; 12Department of Neurosurgery, Technical University of Munich, 80333 Munich, Germany; 13Department of Neurological Surgery, University of Washington, Seattle, WA 98115, USA; 14Department of Radiation Oncology, Technical University of Munich, 81675 Munich, Germany

**Keywords:** spine and sacral chordoma, radiation therapy, stereotactic body radiation therapy, proton therapy, heavy ion therapy, carbon ion therapy

## Abstract

**Simple Summary:**

Chordomas are rare tumors of the embryologic spinal cord remnant. They are locally aggressive and typically managed with surgery in combination with radiation therapy. However, there is great variability in practice patterns including different radiation treatment types and approaches, and limited high-level data to drive decision making. The purpose of this manuscript was to summarize the current literature specific to radiotherapy in the management of spine and sacral chordoma and to provide a practical guide on behalf of the Spine Tumor Academy, an international group of spinal oncology experts.

**Abstract:**

Chordomas are rare tumors of the embryologic spinal cord remnant. They are locally aggressive and typically managed with surgery and either adjuvant or neoadjuvant radiation therapy. However, there is great variability in practice patterns including radiation type and fractionation regimen, and limited high-level data to drive decision making. The purpose of this manuscript was to summarize the current literature specific to radiotherapy in the management of spine and sacral chordoma and to provide practice recommendations on behalf of the Spine Tumor Academy. A systematic review of the literature was performed using the Preferred Reporting Items for Systematic reviews and Meta-Analyses (PRISMA) approach. Medline and Embase databases were utilized. The primary outcome measure was the rate of local control. A detailed review and interpretation of eligible studies is provided in the manuscript tables and text. Recommendations were defined as follows: (1) consensus: approved by >75% of experts; (2) predominant: approved by >50% of experts; (3) controversial: not approved by a majority of experts. Expert consensus supports dose escalation as critical in optimizing local control following radiation therapy for chordoma. In addition, comprehensive target volumes including sites of potential microscopic involvement improve local control compared with focal targets. Level I and high-quality multi-institutional data comparing treatment modalities, sequencing of radiation and surgery, and dose/fractionation schedules are needed to optimize patient outcomes in this locally aggressive malignancy.

## 1. Introduction

Chordomas are rare tumors of the embryologic notochord remnant. They may occur anywhere within the axial skeleton, but are most common in the base of skull or sacrum. However, chordomas do occur in the mobile spine as well. Although pathologically benign in appearance and generally slow growing with a median overall survival of approximately a decade [1], these tumors are considered malignant as they have metastatic potential. Specifically, 5–40% of patients develop distant metastases during their disease course [2]. Nonetheless, the primary cause of morbidity and mortality in chordoma is local recurrence. 

Given the locally aggressive nature of chordoma, the standard-of-care management consists of aggressive surgical resection in combination with either neoadjuvant or adjuvant radiation therapy as deemed clinically appropriate. The role of radiation therapy is controversial and there are no level 1 data to guide decision making. As such, the optimal radiation technique and sequencing remains unclear and may consist of proton, photon, or heavy ion therapy using either conventional fractionation or hypofractionated stereotactic radiosurgery. The purpose of this collaboration was to summarize the current literature specific to radiotherapy in the management of spine and sacral chordoma and to provide practice recommendations for treatment on behalf of the Spine Tumor Academy. A brief summary of imaging and surgical approaches is also included for the benefit of the oncology audience.

## 2. Materials and Methods

A systematic review of the literature was performed using the Preferred Reporting Items for Systematic reviews and Meta-Analyses (PRISMA) approach.

### 2.1. Search Strategy

Medline and Embase databases were utilized to search for manuscripts reporting outcomes following surgery and radiation therapy for spine and sacral chordoma with a search end date of 29 October 2021. Search words included “spine OR spinal OR sacrum OR sacral” AND the following: “chordoma and radiation”, “chordoma and stereotactic”, “chordoma and SRS”, “chordoma and SABR”, “chordoma and SBRT”, “chordoma and radiosurgery”, “chordoma and carbon”, “chordoma and IMRT”, and “chordoma and external beam”. Prospective studies and retrospective series that included at least 10 patients with spinal/sacral chordoma with results specific to the spinal/sacral chordoma subtype reported separately were included. Studies that included skull base chordoma were also included provided that results for the spinal/sacral subgroup were reported separately. Only studies published in English or with an English translation available were considered eligible. Clinicaltrials.gov was also utilized to identify ongoing trials evaluating radiation therapy approaches in spine/sacral chordoma. Abstracts without a published manuscript were excluded, as were dosimetric analyses without clinical outcome data, systematic reviews, meta-analyses, pre-clinical studies, and those in which clinical outcomes were not reported. In addition, manuscripts that reported outcomes for multiple histologies in combination with chordoma, studies including patients who did not undergo radiation or in which details of radiation dose and technique were not available, and manuscripts that reported outcomes for chordomas of skull base in combination with chordoma of the spine/sacrum were excluded from this review.

### 2.2. Outcome Measures

Data collected during the systematic review included local control, tumor location, surgery including extent of surgery and timing relative to RT, radiation technique and modality, prescription dose/fractionation, prior overlapping RT including type and number of patients, and overall survival. Toxicity including but not limited to wound healing complications, spinal cord myelopathy, and nerve plexopathy were included.

The spine tumor academy is an international multi-disciplinary academic collaboration of spinal oncology experts across fields including neurological surgery, orthopedic surgery, radiation oncology, medical oncology and neuro-radiology. A preliminary draft of the manuscript was reviewed at the December 2021 Spine Tumor Academy meeting which was attended by 55 people from six countries including Germany, Canada, the United States, Austria, the Netherlands, and Italy. The manuscript then underwent serial revisions and peer review by members of the Spine Tumor Academy. Ultimately, 15 experts were offered authorship given their leadership roles and extensive contributions to the manuscript. Levels of agreement regarding the recommendations outlined in the guidelines were defined as follows: (1) consensus: selected by at least 75% of respondents; (2) predominant: selected by at least 50% of respondents; and (3) controversial: no single response selected by a majority of respondents. Descriptive statistics were used to review the results.

## 3. Results

The details of the PRISMA search are shown in Figure 1. 

Primary database screening identified a total of 1215 candidate citations (714 from Embase and 501 from Medline). After removal of 439 duplicates, 173 conference abstracts, 80 review articles, 14 commentary, 12 letters, 6 editorials, 5 conference reviews, 4 short surveys, and 1 erratum, 481 candidate citations remained. Of those 481, 45 met the inclusion criteria, including those reporting clinical outcomes of ≥10 patients with chordoma of the spine/sacrum treated with radiation.

### 3.1. Proton Beam Therapy

Proton beam therapy (PBT) is a charged particle-based treatment that has been shown to address the need for dose escalation to the target for improved tumor control with the ability to spare critical organs at risk (OAR). This is achieved by the intrinsic physical properties of proton therapy where there is a penetrating dose deposition along the beam path as the particle slows down until it stops at the end of the range at which it deposits most of its dose, described by the characteristic Bragg peak, with no exit dose. Compared with photon-based radiotherapy with an exponential decay function, this allows for reduced dose to OARs distal to the desired target and decreased integral dose (low-dose bath) of radiation that may translate into reduced acute and late RT treatment morbidity and secondary malignancy risk.

Evolution in the technology for delivery of proton therapy from passive-scattered (e.g., double-scatter) to pencil beam scanning (PBS) has allowed for increased high-dose conformality, particularly for the proximal component of the target, such as with concave target volumes (e.g., chordomas involving the vertebral bodies that require sparing of the adjacent spinal cord) and decreased skin dose. PBS consists of a thin pencil-beam “spot” that has a given depth defined by the beam energy. This “spot” is actively scanned with magnets on a voxel-basis on a given layer. Then with modulation of the beam energy, dose painting of the next layer commences until the target coverage is complete. 

Beam angle selection is of paramount importance for PBT to maximize target coverage robustness and minimize range uncertainty. Key considerations in regard to beam selection for chordoma proton therapy plans include the following: (1) limiting distance from entrance to the target; (2) minimizing the entry beam path traversing structures with air and/or bowel gas with uncertain positions on a daily basis; (3) limiting beam number to reduce integral dose; (4) maximizing beam angle separation for maximal skin sparing (e.g., this may require prone positioning of the patient to avoid rails on the table for lower T, L spine, and sacrum plans); (5) avoiding multiple beams’ end of ranges occurring in the same structure, particularly neural structures, given concern for increased relative biological effectiveness; and (6) if high Z surgical stabilization hardware is present (e.g., titanium), minimizing traversing through hardware with consideration of non-coplanar beams and/or mixed photon/proton treatment plans to maximize confidence in dose-delivery to the target and improve confidence in critical OAR dosimetry. PBT treatment plans for chordomas below the spinal cord often consist of two posterior oblique beams separated at an optimal angle for maximal skin sparing and robustness, while plans at the level of the spinal cord in the mobile spine (typically L1–2 and above) may require up to 4–6 different angles depending on the location of the tumor in relationship to the spinal cord, plexus, and other critical OARs. 

PBT doses are expressed as GyRBE (relative biological equivalent) with a conversion factor of 1.1 used to account for its higher relative biological properties. Most studies evaluating PBT have investigated dose escalation to total doses ≥70 GyRBE in conventional fractionation (1.8–2 Gy per fraction), daily, five times per week. In general, comprehensive target volume coverage particularly for the at-risk microscopic clinical target volume (CTV) has been employed, which contrasts with reported more focal target volumes typically used with other heavy particle therapy and SBRT. For spinal cord delineation, the Massachusetts General Hospital (MGH) [3,4] and Paul Scherrer Institute (PSI) [5] method is the most well described, where the treatment planning CT is fused to a T2 MRI or CT myelogram, if surgical hardware is present, to delineate the spinal cord into two structures: (1) spinal cord core (cSC) which is a 2–3 mm region-of-interest in the geometric center of the spinal cord; and (2) the spinal cord surface (sSC).

Our systematic review identified two prospective [3,4,6] and 21 retrospective manuscripts [5,7,8,9,10,11,12,13,14,15,16,17,18,19,20,21,22,23,24,25] that met the inclusion criteria for evaluating outcomes for primary and recurrent mobile spine and/or sacral chordomas treated with PBT in the preoperative/postoperative, adjuvant, and/or definitive setting. Table 1 (Data summary of studies reporting outcomes of patients with spinal and sacral chordoma treated with proton irradiation) summarizes the PBT studies. With a median follow-up of 47 months (range 12.9–87.6 months) across all studies, median overall 5-year local-progression-free survival was 73.3%, ranging from 53% to 85.4%, with median crude local failure rates of 30% (range 13.7–38.8%). Local failure was the dominant pattern of failure with lower rates of developing distant metastatic disease (median 15.5%, range 7–29%). Median time to local failure was 24 months but can often occur years from PBT (range 1–146 months) with one study showing 35% of local failures occurring after 5 years [18]. This highlights the need for caution when interpreting promising early results of outcomes from series with limited follow-up. Median 5-year overall survival among reported studies was 81.3% (range 50% to 100%). 

Common themes emerged regarding adverse prognostic factors of local control for patients treated with PBT. Treatment in the upfront setting for primary chordomas resulted in more optimal outcomes compared with treatment in the recurrent setting [4,7,10,15,19,24], reiterating the importance of upfront multi-disciplinary evaluation for timely and appropriate multimodal care. Given nearly all PBT series included only patients treated with dose-escalated radiation therapy (≥60–70 Gy or higher), a clear a dose–response relationship was not identified except for in one study of sacral chordomas showing improved local control with doses ≥ 70 Gy (HR 0.52, *p* = 0.17) particularly amongst patients with an R1 margin (HR 0.40, *p* = 0.051) or those treated with PBT compared with photon therapy (HR 0.56, *p* = 0.23) [11]. 

High Z surgical stabilization hardware (e.g., titanium) raises concern for technical limitations and dosimetric uncertainty that may contribute to dose “shadowing” (under-dosage) of the target distal to the beam path with particle therapy, where the experience from PSI reports a significant decrement in the 5-year local control of 73.4% and 50% for patients without and with surgical stabilization, respectively (*p* = 0.02) [20,22]. Potential solutions to mitigate this effect include upfront evaluation with the surgeon to determine the extent of surgery indicated and/or necessity of hardware, position of hardware, the consideration of novel carbon-reinforced polyetheretherketone (PEEK) stabilization alternatives that result in reduced CT artifacts and less impact on proton dosimetry because of lower Z composition [26,27], evaluating the feasibility of a mixed photon/proton plan, and/or delivering a meaningful component of the microscopic dose (e.g., 19.8–50.4 GyRBE) in the preoperative setting prior to a postoperative boost to reduce the need to cover the entire surgical resection bed. Importantly, MGH has shown that using a preoperative followed by an individualized post-operative boost approach compared with adjuvant PBT alone results in improved 5-year local control of 85% vs. 56%, respectively (*p* = 0.019), with no local failures for patients who underwent en bloc resection [19]. 

There is a clinical need for consensus guidelines regarding target and critical OAR delineation and dose constraints for chordoma patients treated with PBT, as some series with more focal target volumes suggest inferior local control compared with more comprehensive volumes [20,21,22,28] as well as more frequent patterns of failure in proximity to dose-limiting OARs, such as the spinal cord [20]. 

While PBT allows for decreased dose to OARs distal to the target, critical structures immediately adjacent or within the target volume are still at risk for significant treatment-related morbidity because of the high doses required for tumor control. Across all PBT series, there were only two incidences of grade 3 or greater spinal cord myelopathy, where one patient developed renewed tetraplegia 17 months after initially presenting with temporary tetraparesis that improved with surgical decompression (Dmax to sSC and cSC were 57.8 GyRBE and 54.1 Gy RBE, respectively) [5,22], and the second patient developed transient paralysis 2 years after treatment when undergoing chemotherapy conditioning for an autologous stem cell transplant for myelodysplastic syndrome [16]. In the subacute setting, there is a reported approximate 5% rate of Lhermitte’s syndrome, which is a temporary demyelination phenomenon that resolves spontaneously [5,24]. In the PSI series, when adhering to dose constraints of D2% of the sSC receiving 64 GyRBE (reduced to 60 GyRBE if the target volume was longer than 3 vertebrae) and the cSC receiving 54 GyRBE, only 4% (*n* = 3/71) developed grade 2 or greater neurologic toxicity, whereas 40% (*n* = 2/5) whose dose constraints were exceeded developed toxicity [5]. Nerve plexus neuropathies have been reported in approximately 3–5%, which may manifest as pain, numbness, tingling, weakness, foot drop, erectile dysfunction, and bladder or bowel dysfunction, where doses are typically in the range of 77.4–85 GyRBE when they have been reported [4]. 

Other toxicities after PBT include a significant impact on the rate of wound healing toxicity with reported values of 21.6% (predominantly in patients with sacral tumors) treated with preoperative PBT compared with 12% for those treated with postoperative PBT alone [19], highlighting the absence of “skin sparing” with proton therapy and the importance of close collaboration with surgical colleagues, including plastic surgery, for consideration of flap-based closures to maximize wound healing. Other reported adverse events include a low (0–5%) rate of insufficiency fracture, esophageal stricture, subcutaneous fistula, femoral insufficiency requiring hip replacement, ureteral stenosis, laryngeal necrosis, rectal ulcer and bleeding, menopause, and bowel fistula or perforation requiring a colostomy. For sacral chordomas within 1 cm of the small bowel and/or rectum, one may consider upfront surgical spacer placement, which allows for the necessary distance for particle beam dose fall-off [23]. Reported rates of secondary malignancy are 0–5%.

### 3.2. Carbon Ion and Other Heavy Particle Therapy

There is an ongoing discussion about the efficacy of carbon ion radiotherapy in chordomas. In relation to protons, carbon ions offer comparable physical properties, with a low energy (and thus dose) deposition in the entry channel of the beam and precise dose deposition in the Bragg Peak, followed by a steep dose fall-off in normal tissue behind the target [29]. This, as in protons, leads to a reduction of integral dose in patients. In contrast to protons and photons, carbon ions are associated with a higher relative biological effectiveness (RBE); several preclinical studies have demonstrated this increased efficacy in various tumor entities, including pancreatic cancer, gliomas, and also sarcomas [30,31,32,33,34,35,36,37,38]. Moreover, there is a strong rationale that carbon ions can overcome radiation resistance caused by hypoxia [38]. Since chordomas are radiation-resistant tumors requiring high local doses, there is a strong rationale for carbon ions in this tumor entity, not only in terms of dose escalation based on the superior dose distribution of particles, but also based on the biological properties. 

However, to date, no large series are available for chordomas of the mobile spine. Regarding chordomas, most data are available from skull-base chordomas, where particle therapy probably has the strongest rational especially because of the intricate anatomy. Most large series based on skull-base chordomas report local control rates that are relatively high compared with older photon series. For example, Koto et al. reported on 34 patients treated with 60.8 Gy E in 16 fractions and demonstrated local control of 76.9% at 5 years and 69.2% at 9 years [39]. A recent Heidelberg series by Uhl and colleagues including 155 skull-base chordomas treated with carbon ions published a local control rate of 72% and 54% at 5 and 10 years [40]. For chordomas located along the mobile spine, the data are scarce; however, smaller series have demonstrated high efficacy and low rates of side effects in a number of tumor entities and locations. The data are often mixed with chondrosarcomas of the spine, or analyzed together with sacral chordomas which are generally a different entity because of the surgical and also radiation oncology requirements related to the distinct differences in anatomy. 

In terms of toxicity, rates of sacral fractures following carbon ion therapy for sacral chordoma were high, impacting approximately half of patients [41]. However, the authors did note that only about a third of fractures were clinically symptomatic, requiring regular medical care and pain therapy. In addition, rates of wound healing complications following carbon ion and heavy particle therapy were high. For example, a study of patients treated with helium and neon therapy demonstrated a 35% rate of chronic wound complications [42,43].

Table 2 summarizes the eligible series of carbon ion radiotherapy including chordomas of the mobile spine. The readers are also directed to a comprehensive review of spinal and sacral chordomas treated with carbon ions written by Pennington et al. [44].

### 3.3. Stereotactic Body Radiation Therapy (SBRT)

Advances in radiation technology including micro-multileaf collimators, cone beam CT scans, robotic systems, and real-time image guidance have allowed for progressively more precise delivery of photon therapy utilizing steep dose gradients and the emergence of SBRT. SBRT is increasingly available at many community and academic centers throughout the world, and thus is more readily available than charged particle therapies such as proton and carbon ion therapy, which have been discussed in earlier sections. Hypofractionated stereotactic regimens allow the delivery of ablative doses of radiation therapy by limiting the dose to adjacent normal tissues. Compared with conventionally fractionated radiation therapy, SBRT activates unique cell-killing pathways including apoptosis and takes advantage of radiobiologic principles including a decrease in sublethal damage repair and repopulation of tumor cells between fractions. These regimens also help to destroy microvasculature and overcome the traditional radioresistance of hypoxic cells which may be found in the center of large tumors such as chordoma. 

Our systematic review identified nine retrospective manuscripts including a total of 197 patients and no prospective clinical trials that met inclusion criteria for evaluating outcomes for primary and recurrent mobile spine and/or sacral chordomas treated with SBRT in the preoperative/postoperative, adjuvant, and/or definitive setting. The data are shown in Table 3. With a median follow-up of 34 months (range 1.7–216 months) across all studies, the median overall crude local recurrence free survival was 71%, ranging from 45% to 95%. For the series that reported local control for the treatment naïve patients separately, the median overall local recurrence free survival was 92% (range 86–95%). Local failure was the primary pattern of failure with distant metastatic disease developing in a median of 17.5% (range 0–30%). The median crude overall survival among reported studies was 72% (range 59.3% to 92%). 

Higher prescription doses were reported to be associated with superior local control. Specifically, in a series reporting outcomes of primarily fractionated SBRT, no local recurrences occurred in patients receiving a BED2 < 140 Gy [61]. Similarly, in two series delivering predominantly single fraction SBRT, local control was reported to be approximately 95% in patients receiving 24 Gy [55,60]. Furthermore, superior local control was reported in treatment-naïve patients undergoing definitive management than in the salvage setting. In this light, Chen and colleagues report no local recurrences in their subset of 17 patients receiving neoadjuvant high-dose hypofractionated SBRT followed by surgical resection with curative intent [61].

Manuscripts variably reported the normal tissue constraints that were utilized in treatment planning. Of those that reported spinal cord constraints, 14 Gy in a single fraction was used for the true spinal cord in a single fraction and 25.3 Gy was used for the spinal cord plus 2 mm or thecal sac in five fractions.

Overall, the toxicities associated with SBRT were low and generally correlated with anticipated sequelae based on the treated spinal levels. Skin toxicity was rare and although there was no direct comparison of modalities across studies, seemingly lower than that reported in studies of heavy particle therapy. Of greater concern, a single study [57] did report at 13% risk of grade 2 spinal cord myelopathy following SBRT, although it did resolve with corticosteroid administration. 

A significant concern with adjuvant and neoadjuvant radiation therapy is the risk of wound healing complications; however, the rates reported were low in a median of 8.9% of patients (range: 3.3–1%) in studies that reported this toxicity [58,61,62]. This rate is comparable to patients undergoing surgery alone and lower than observed in patients treated with alternative approaches such as proton therapy. We speculate that this may be because of lower skin doses with SBRT given the rapid radiation dose fall-off over millimeters. 

Although aggressive surgical resection remains the cornerstone of care, emerging data from a single institution suggest reasonable local control with SBRT alone. Specifically, Yamada and colleagues [55] reported 95% local control following 24 Gy in a single fraction of SBRT in a cohort of patients in which nearly one-third did not undergo surgery and all surgical patients had gross residual disease post-operatively. However, it is critical to note that the median follow-up in this series was only 38.8 months overall and 16.5 months in the subset of treatment-naïve patients. It is possible that these control rates may decrease over time and long-term follow-up is necessary.

### 3.4. Radiation Alone

Although aggressive surgical resection in considered the standard of care in the management of chordoma, there are times when radiation alone may be considered. The primary advantage of definitive radiotherapy is a reduction in morbidity and recovery from surgery or as a management option in medically inoperable patients. To date, there are no prospective or retrospective studies comparing radiation therapy alone to surgery followed by radiation therapy or comparing radiation treatment modalities in patients undergoing radiation alone. Our systematic review identified nine retrospective manuscripts and a single prospective phase 1–2 clinical trial including a total of 641 patients treated with radiation alone that met the inclusion criteria for evaluating outcomes for primary and recurrent mobile spine and/or sacral chordomas treated with radiation therapy alone. These data are summarized in Table 4. With a median follow-up of 52 months (range 37–80 months) across all studies, the median local control (at 3–5 year depending on the study) was 80%, ranging from 62% to 94%. Four studies utilized exclusively carbon ion therapy, two exclusively proton therapy, and three utilized combinations of charged particle ± photon therapies. For these fractionated regimens, four had a median prescription dose of 70.4 Gy RBE while three others had higher median prescription doses ranging from 74–80 Gy RBE in fractions ranging from 2.2–4.6 Gy RBE. A single study reported outcomes following SBRT alone to a median dose of 24 Gy in a single fraction and revealed a 2-year local control of 100%. The median 5-year overall survival for the studies was reported as 84% (range 74% to 88%). Toxicities were limited with the most common sequelae including sacral insufficiency fractures as well as both acute and late skin complications.

Taken in aggregate, radiation alone remains a reasonable option in a subset of patients who are medically inoperable or elect to forgo the potential risks associated with an aggressive surgical procedure. It is important to note that all studies utilized relatively dose-escalated prescription doses in an effort to overcome the known radioresistance of chordoma. Although there are currently not sufficient data to compare outcomes following protons, carbon ion, photon, and SBRT, the local control across studies was excellent, although a longer-term follow-up will be essential.

### 3.5. Timing of RT

Local recurrence or progression following surgical resection occurs frequently because of the inability to achieve wide margin excision in patients with spinal and sacral chordomas. Postoperative radiotherapy using the approaches described above including photon-based intensity modulated radiotherapy, proton therapy, and carbon ion therapy has been utilized to improve local control. Based on sarcoma literature, delivery of radiation therapy in the adjuvant setting may minimize the risk of wound healing complications. In addition, it allows providers to determine the need for RT based on the extent of resection and to work with surgeons to identify the regions of close or positive margins that may be at the highest risk of recurrence. However, target delineation is more challenging in the adjuvant setting given the difficulty in discerning post-operative change from residual/recurrent disease. In addition, it is possible that tumor cells may contaminate the surgical field at areas more remote from the original gross disease. As a result, the radiation target is typically larger in the adjuvant setting than in the neoadjuvant setting, resulting in the delivery of higher doses of radiation to adjacent normal tissues. 

By contrast, neoadjuvant radiotherapy simplifies target delineation as the characteristic T2 hyperintense regions of gross disease may be identified with greater confidence than in the post-operative setting. As a result, the margins may be tighter, minimizing radiation dose to the adjacent normal structure. Discussions with surgeons are essential to identify the regions at highest risk of a positive margin post-operatively so that the target and prescription dose may be modified accordingly. The rationale of neoadjuvant radiation therapy is to effectively sterilize any cells that may spill from the capsule at the time of surgery and thereby reduce the risk of microscopic residual leading to local recurrence. In addition, it may be beneficial to facilitate negative margins but with less surgical morbidity associated with sacrifice of critical neural structures. The greatest concern in this setting is the potentially increased risk of wound healing complications. To minimize this risk, care must be taken to minimize radiation dose to the skin, especially since a clear superficial margin is generally not a challenge at the time of surgical resection. 

Ultimately, the decision to offer radiation therapy in the adjuvant or neoadjuvant settings is often driven by institutional bias, as there have been no studies directly comparing the two approaches. Some institutions utilize a compromise approach and deliver some dose in the neoadjuvant setting with and additional boost post-operatively. 

Several retrospective studies have attempted to compare outcomes based on the timing of radiation therapy. For example, in a study from MGH [19], patients were treated with either adjuvant radiation therapy using photons or a combination of pre-operative plus postoperative combined photon and proton therapy. Patients who had preoperative plus postoperative radiotherapy showed a trend toward superior local control. However, an alternative retrospective study by the Sacral Tumor Society [11] suggests increased risk of wound complications using this approach.

Other studies have evaluated SBRT in the adjuvant and neoadjuvant setting. For example, the Johns Hopkins University [61] series demonstrated negative margins in all patients undergoing en bloc resection following neoadjuvant SBRT with no local recurrences during the study period. It is important to note that approximately one-third of patients developed post-operative wound healing complications, although the authors noted that this rate is comparable to the rate in patients undergoing surgery alone without radiation therapy. Memorial Sloan-Kettering Cancer Center [60] similarly reported excellent outcomes in 11 sacral chordoma patients receiving preoperative SBRT, with a 3-year local recurrence-free survival of 90%. However, they did not report outcomes specific to the adjuvant radiotherapy group and complications were not reported separately based on the timing of radiation. As such, given the very limited literature, the optimal timing of radiation therapy relative to surgery remains unclear. Table 5 summarizes studies showing outcomes for both preoperative and postoperative RT for mobile spine/sacral chordomas.

### 3.6. Summary of Ongoing Clinical Trials 

Given the relatively high recurrence rates in management of chordoma, clinical trials are of the utmost importance in improving outcomes and optimizing management. Table 6 summarizes the 16 ongoing and completed but not published clinical trials involving radiotherapy for spinal and sacral chordoma that were listed on clinicaltrials.gov on the search completion date of 29 October 2021. The most common subject is an evaluation of efficacy and/or toxicity of proton therapy either alone or in combination with surgery. There are two studies incorporating PET imaging to identify hypoxic cells in target delineation for proton therapy. Although surgery remains the gold standard, SACRO is a randomized controlled trial that is currently accruing in Italy which is randomizing patients with sacral chordoma to definitive RT versus surgery. The results of this exploration will be critical given the high morbidity of en bloc sacrectomy, which may be avoided with definitive RT alone. There are three studies comparing outcomes of carbon ion therapy with proton therapy. Finally, three additional studies are exploring the addition of novel systemic therapies including nilotinib, nivolumab and brachyurea to radiotherapy. Taken together, this important compendium of studies will help advance the field in our understanding of the optimal radiation technique and help explore mechanisms to improve outcomes in this rare and aggressive malignancy. 

### 3.7. Summary of Radiotherapy Recommendations 

Overall consensus recommendations from the Spine Tumor Academy are shown in Table 7. 

### 3.8. Limitations

Only 45 of 481 candidate citations met the inclusion criteria and were deemed eligible for inclusion in this systematic review. In addition, only two of the included studies were prospective in nature. Therefore, the preponderance of data driving these guidelines are taken from small retrospective studies that variably reported specific outcomes. As such, they suffer from challenges characteristic of single-institution and retrospective series including patients lost to follow-up, reporting bias, and selection bias. In addition, many of these studies have short follow-up periods of only a few years, which is particularly challenging given the protracted disease course of patients with chordoma. Specifically, it is unclear whether optimistic local control outcomes at short intervals will translate into similarly strong outcomes in the ensuing decade(s) that a chordoma patient would be predicted to live. Ultimately caution must be utilized when interpreting promising early results of outcomes from series with limited follow-up.

Furthermore, the included studies generally did not compare dose fractionation regimens, treatment modality of treatment, or timing for radiation therapy. Although we are able to draw conclusions that certain modalities may trend to higher (or lower) rates of certain toxicities and surmise that high doses of radiation are essential for local control of mobile spine and sacral chordoma, level I data comparing different approaches are unavailable. Therefore, although we present practice recommendations that have been heavily reviewed and discussed amongst multi-disciplinary experts in the international Spine Tumor Academy based on the best available data, larger-scale and multi-institutional studies (such as those under development by the Spine Tumor Academy) will be essential in optimizing patient outcomes in this locally aggressive malignancy. It should be noted that the literature search for this manuscript included primary research citations rather than previously published review articles. Nonetheless, a few additional reviews on this topic have been published in the last decade and are referenced here for the interested reader to access if desired [44,65,66,67].

## 4. Conclusions

To conclude, multi-disciplinary expert involvement at the time of initial diagnosis of Kmobile spine and sacral chordoma is critical to optimizing patient outcomes. Although high-level data comparing outcomes, dose/fractionation regimens, and treatment modalities are unavailable, dose escalation is critical in optimizing local control. Target delineation should be performed using a CT scan with at minimum a co-registered T2 weighted MRI and should include a careful discussion between the spine surgeon and radiation oncologist. Comprehensive target volumes including sites of potential microscopic involvement improve local control compared with focal targets. Reasonable dose/fractionation schedules by treatment modality include 75.6–77.4 Gy RBE in 1.8–2 Gy RBE fractions using proton ± photon therapy; 24 Gy in a single fraction or 40–50 Gy in five fractions of SBRT; and at least 70.4 Gy in 2.2–4.4 Gy RBE fractions using carbon ion therapy. In addition, efforts must be made to limit skin dose when using proton therapy and heavy particles to minimize the risk of chronic wound healing complications. Level I and high-quality multi-institutional data comparing treatment modalities, sequencing of radiation and surgery, and dose/fractionation schedules are needed to optimize patient outcomes in this locally aggressive malignancy. 

## Figures and Tables

**Figure 1 cancers-15-02359-f001:**
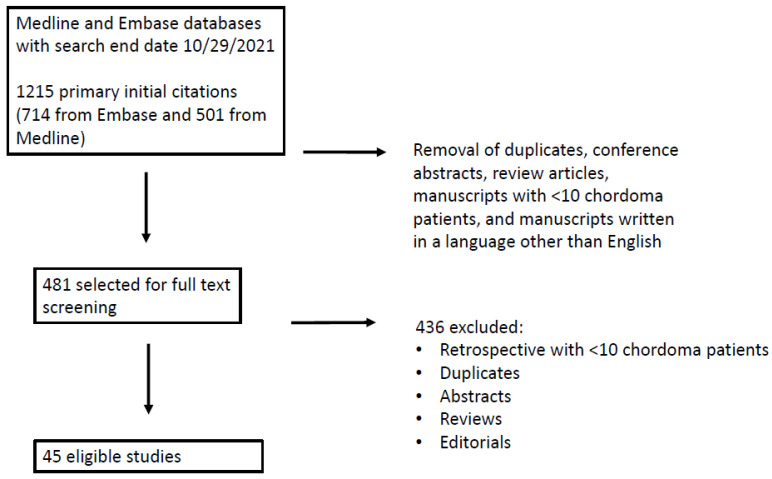
Details of the PRISMA search.

**Table 1 cancers-15-02359-t001:** Data summary of studies reporting outcomes of patients with spinal and sacral chordomas treated with proton irradiation.

Author, Journal, Year Published	Study Type	Number of Patients	Median Follow-Up (mo)	Extent of Resection	Radiation Timing	Prescription Dose (Range)/Dose per Fraction	Local Control	Overall Survival	Toxicity
Austin, IJROBP, 1993 [12]	Retrospective	26	NR	Biopsy, STR, GTR	RT alone, adjuvant	Gross disease—70 Gy RBE/1.8–2 Gy RBE fractions Microscopic disease—45–50 Gy	Crude 62%	NR	NR
Fagundes, IJROBP, 1995 [13]	Retrospective	69	39	STR, GTR	Adjuvant	Median 70.1 Gy RBE (66.6–77.4)	Crude 65%	NR	NR
Hug, IJROBP, 1995 [10]	Retrospective	14 ^‡^	38	Biopsy, STR, GTR	RT alone, pre/postoperative, adjuvant	Mean 74.6 Gy RBE (67.1–82)/1.8–2 Gy fractions	5 yr 53%	5 yr 50%	6% attributable to RT
Park, IJROBP, 2006 [15]	Retrospective	27	47	Biopsy, STR, GTR	RT alone, adjuvant	Primary—Mean 71 Gy RBE/1.97 Gy RBE fractions Recurrent—Mean 77 Gy RBE/1.88 Gy RBE fractions	5 yr 71.7%	5 yr 82.5%	37% abnormal bowel function; 30% pain; 19% abnormal bladder function; 11% difficulty ambulating
Wagner, IJROBP, 2009 [7]	Retrospective	25	32	STR, GTR	Pre/postoperative, adjuvant	Preoperative—Median 20 Gy RBE (9–29.4) Postoperative—Median 50.4 Gy RBE (18–61.2)	5 yr 73.3%	5 yr 65%	21% delayed wound healing; 11% late toxicity
Staab, IJROBP, 2011 [24]	Retrospective	40	NR	Biopsy, STR, GTR	RT alone, adjuvant	Mean 72.5 Gy RBE (59.4–75.2)/1.8–2 Gy RBE fractions (93% received ≥ 70 Gy RBE)	5 yr 62%	5 yr 80%	4% G3 osteonecrosis, 4% subcutaneous fistula requiring wound debridement; 0% G3 neuro, kidney, and bowel toxicity
Chen, Spine, 2013 [8]	Retrospective	24	56	Biopsy	RT alone	Median 77.4 Gy (70.2–79)/1.8–2 Gy RBE fractions Median photon contribution 34 Gy (0–57.6) Median proton contribution 45 Gy RBE (9.8–79.2)	5 yr 79.8%	5 yr 78.1%	33% sacral insufficiency fracture (none requiring stabilization); 17% G2 rectal bleeding; 8% worsening fecal/urine incontinence; 4% foot drop; 4% perineal numbness; 4% erectile dysfunction; 1% secondary malignancy
Kim, Acta Oncol, 2014 [25]	Retrospective	12	43	STR, GTR	RT alone, adjuvant	Median NR (64.8–79.2)/2.4 Gy RBE fractions	Crude 83%	NR	17% G3 skin/subcutaneous contracture; 8% G3 rectal bleeding
Delaney, J Surg Onc, 2014 [4]	Prospective phase II	29	88 (among alive patients)	Biopsy, STR, GTR	RT alone, pre/postoperative, adjuvant	Median 76.6 Gy (59.4–77.4)/1.8–2 Gy RBE fractions	5 yr 81%	5 yr 84%	13% 8 yr actuarial risk of G3–G4 late RT morbidity; 3 sacral neuropathies (all after doses of 76.6–77.4 Gy); no myelopathies
Rotondo, J Neurosurg Spine, 2015 [19]	Retrospective	126	47	Biopsy, STR, GTR	RT alone, pre/postoperative, adjuvant	Median 72.4 Gy RBE (46.3–83.6)/1.8–2 Gy RBE fractions	5 yr 62%	5 yr 81%	22% wound complications with preoperative RT, 12% with postoperative RT; 5% insufficiency fracture; 3% motor neuropathy; 2% spine non-union/hardware failure; 1% secondary malignancy; 1% proctitis; 1% osteonecrosis; 1% erectile dysfunction
Indelicato, IJROBP, 2016 [6]	Retrospective	34	44	Biopsy, STR, GTR	RT alone, adjuvant	CTV + 5 mm–Median 45 Gy RBE/1.8–2 Gy RBE fractions GTV + 5 mm–Median 70.2 Gy RBE (65–75)/1.8–2 Gy RBE fractions daily or 1.2 Gy RBE fractions BID	4 yr 67%	4 yr 72%	5% G3–G4 soft tissue toxicity/wound healing; 5% secondary malignancy; 2% compression fracture requiring stabilization; 2% bilateral G2 radiation nephritis
Chowdhry, IJROBP, 2016 [16]	Retrospective	29	13	NR	Pre/postoperative, adjuvant	Preoperative—Median 36 Gy RBE (18–78.2)/1.8–2 Gy RBE fractions Adjuvant—Median 70.2 Gy RBE (59.4–78.2)/1.8–2 Gy RBE fractions Photon contribution—19.8–30.6 Gy Proton contribution—Remaining dose	NR	5 yr 86.9%	7% G2+ neurologic injury
Kabolizadeh, IJROBP, 2017 [17]	Retrospective	40	50	Biopsy	RT alone	Median 77.4 Gy RBE (64.8–79.2) Photon contribution—Median 30.6 Gy (0–68) Proton contribution—Median 46.8 Gy RBE (0–79.2)	5 yr 85.4%	5 yr 81.9%	25% sacral stress fracture (none requiring surgical stabilization); 10% G2 rectal bleeding; 5% urinary/fecal incontinence; 5% secondary malignancy; 5% foot drop; 3% erectile dysfunction; 3% perineal numbness; 3% bowel fistula/perforation
Stieb, IJROBP, 2018 [5]	Retrospective	55	66	Biopsy, STR, GTR	RT alone, adjuvant	Median 73.9 Gy RBE (59–75.2)/1.8–2 Gy RBE fractions	5 yr 61%	Median 65 mo 5 yr 75%	5% acute RT-induced neurotoxicity (1% G1, 4% G2); 16% late neurologic toxicity (9% G1, 5% G2, 1% G4)
Snider, IJROBP, 2018 [20]	Retrospective	100	65	Biopsy	RT alone, adjuvant	Median 74 Gy RBE (59.4–77)/1.8–2 Gy RBE fractions (95% received ≥ 70 Gy)	5 yr 63%	Median 157 mo 5 yr 81%	11% G3
Aibe, IJROBP, 2018 [21]	Retrospective	33	37	Biopsy	RT alone	70.4 Gy RBE/2.2 Gy RBE fractions	3 yr 89.6%	3 yr 92.7%	13% acute G3; 3% leg numbness
Tsugawa, J Am Coll Surg, 2020 [23]	Retrospective	21	50	Biopsy	RT alone	Early treatment era—70.4 Gy RBE/4.4 Gy RBE fractions Modern era—70.4 Gy RBE/2.2 Gy RBE fractions	4 yr 68.4%	5 yr 100%	19% acute G3 dermatitis; 5% late G4 dermatitis
Houdek, J Surg Onc, 2019 [11]	Retrospective	89	84	STR, GTR	Pre/postoperative, neoadjuvant, adjuvant	Pre/postoperative—Mean 70.9 Gy RBE (±5.7) Neoadjuvant—50 Gy RBE Adjuvant—Mean 60.2 Gy RBE (±9.9)	5 yr 80%	Median 60 mo	39% wound dehiscence/delayed healing; 20% sacral stress fracture; 3% secondary malignancy; 3% small bowel obstruction; 1% enteric fistula
Fujiwara, Int Orthop, 2020 [14]	Retrospective	11	77	GTR	Adjuvant	NR	5 yr 82%	NR	NR
Murray, J Neurosurg Spine, 2020 [22]	Retrospective	116	65	STR, GTR	Adjuvant	Median 74 Gy RBE (59.4–77)	5 yr 67.9%	5 yr 81.6%	34% long-term RT-induced toxicity: 7% G3, 1% G4 (myelitis causing quadriplegia, laryngeal necrosis requiring hyperbaric oxygen)
Beddok, Acta Oncologica, 2021 [9]	Retrospective	28	34	Biopsy, STR	RT alone, adjuvant	CTV—Median NR (52.2–54 Gy RBE)/1.8–2 Gy RBE fractions GTV + 5 mm–Median NR (70–73.8 Gy RBE)/1.8–2 Gy RBE fractions	5 yr 75%	5 yr 74.5%	14% G2 & 4% G3 late pain; 4% G2 late fibrosis; 9% G2 late cauda equina syndrome
Walser, Clinical Oncology, 2021 [18]	Retrospective	60	48	Biopsy, STR, GTR	RT alone, adjuvant	Median 74 Gy RBE (60–77)/1.8–3 Gy RBE fractions	4 yr 77%	4 yr 85%	Acute: 43% G2, 10% G3 Late: 30% G2, 5% G3 (3% sacral insufficiency fracture, 1% neuropathic pain interfering with ADL), 3% G4–G5 (secondary malignancy)

Abbreviations: mo = months; NR = not reported; STR = subtotal resection; GTR = gross total resection; RT = radiation therapy; yr = year; G3 = grade 3; G2 = grade 2; G4 = grade 4; CTV = clinical target volume; GTV = gross tumor volume; BID = twice daily; G1 = grade 1; ADL = activities of daily living; G5 = grade 5. Mean provided when median not reported. Data are listed for the specific group when available or the overall cohort if group-specific data are not available.

**Table 2 cancers-15-02359-t002:** Data summary of studies reporting outcomes of patients with spinal and sacral chordomas treated with carbon ion and other heavy particle therapy.

Author, Journal, Year Published	Study Type	Number of Patients	Median Follow-Up (mo)	Extent of Resection	Radiation Type and Timing	Prescription Dose (Range)/Dose per Fraction	Local Control	Overall Survival	Toxicity
Mima, Br J Radiol, 2014 [28]	Retrospective	23	38	Biopsy	Carbon ion or proton alone	70.4 Gy RBE/2.2 or 4.4 Gy RBE fractions	3 yr 94%	3 yr 83%	39% grade 3 or greater acute 22% late grade 4 dermatitis; 17% grade 3 neuropathies; 9% grade 3 myositis
Uhl, Strahlenther Onkol, 2015 [45]	Retrospective	56	25	Biopsy, STR, GTR	Carbon ion alone or adjuvant ± photon	Median 66 Gy RBE (range 60–74)/3 Gy RBE fractions	3 yr 53%	100%	0% new grade 3 or greater toxicity
Imai, IJROBP, 2016 [46]	Retrospective	188	62	Biopsy	Carbon ion alone	Mean 67.2 Gy RBE (64–73.6)/4–4.6 Gy RBE fractions	5 yr 77.2%	5 yr 81.1%	3% grade 3 neuropathies; 1% grade 4 skin toxicity
Imai, Br J Radiol, 2011 [47]	Retrospective	84	42	Definitive	Carbon ion	Median 70.4 Gy RBE (52.8–73.6)/3.3–4.6 Gy RBE fractions	5 yr 86%	5 yr 88%	2% skin or soft tissue complications requiring skin graft; 16% severe sciatic nerve complications requiring medication
Demizu, Radiat Oncol, 2021 [48]	Retrospective	219	56	Definitive	Carbon ion	67.2 Gy RBE, 70.4 Gy RBE, 79.2 Gy RBE/2.2–4.4 Gy RBE fractions	5 yr 72% *	5 yr 84% *	1.4% grade 3 myositis; 1% insufficiency fracture; 1% skin disorders; 1% tissue necrosis; 2% grade 4 skin disorders
Evangelisti, Eur Rev Med Pharmacol Sci, 2019 [49]	Prospective	18	23.3	Biopsy	Carbon ion alone	70.4 Gy RBE/4.4 Gy RBE fractions	2 yr 84.6%	2 yr 100%	44% late neuropathy; 62.5% grade 1 parasthesia; 37.5% grade 2–3 pain; 5.5% grade 2 late gastrointestinal toxicity
Serizawa, J Compt Assist Tomogr, 2009 [50]	Retrospective	34	46	Biopsy, resection (unspecified extent)	Carbon ion alone or salvage	Range 52.8–73.6 Gy RBE, fraction dose not stated	5 yr 93.8%	5 yr 85.4%	NR
Imai, IJROBP, 2010 [51]	Phase 1–2 and 2	30	80	Definitive	Carbon ion	Median 70.4 Gy RBE (52.8–73.6)/3.3–4.6 Gy RBE fractions	5 yr 89%	5 yr 86%	5% skin or soft tissue complications requiring skin graft *
Bostel, Radiat Oncol, 2020 [52]	Retrospective	68	60.3	Biopsy, STR, GTR	Carbon ion alone, salvage	Median 80 Gy RBE (range, 68.8–96 Gy RBE)	5 yr 53%	5 yr 74%	Grade 3 or greater late effects in 21%; Sacral insufficiency fractures in 49% (36% symptomatic); peripheral neuropathy 9%; skin toxicity 9%; intestine 3% *
Preda, Radiother Oncol, 2018 [53]	Retrospective	39	18	Biopsy	Carbon ion alone	70.4 Gy RBE/4.4 Gy RBE fractions	Cumulative 80%	NR	NR
Bostel, Radiother Oncol, 2018 [41]	Retrospective	56	35.5	Biopsy, STR, GTR	Carbon ion +/− photon alone or adjuvant	Median 66 Gy RBE (range, 60–74 Gy RBE)/3 Gy RBE fractions	NR	NR	52% sacral insufficiency fracture
Schoenthaler, IJROBP, 1993 [42]	Retrospective	14	60	Biopsy, STR, GTR	Adjuvant helium and neon	Median dose 75.65 Gy RBE (range, 70–80.5 Gy RBE)/1.8–2.12 Gy RBE fractions	5 yr 62% neon and 34% helium (55% overall)	5 yr 85%	7% colostomy for rectal injury; 7% second malignancy; 35% chronic wound
Breteau, Bull Cancer Radiother, 1996 [43]	Retrospective	12	NR	Biopsy, STR, GTR	Neutrons alone, salvage	three regimens based on tumor size and intent of therapy: (1) 40 Gy photons plus 15–25 neutron Gy (2) Curative 17.6 neutron Gy, 16 fractions (3) Palliative 10 neutron Gy, 12 fractions	4 yr 54%	4 yr 61%	17% moist desquamation; 25% diarrhea

Abbreviations: mo = months; STR = subtotal resection; GTR = gross total resection; NR = not reported; RT = radiation therapy; yr = year; SBRT = stereotactic body radiation therapy; G1 = grade 1; G2 = grade 2; CI = confidence interval; GI = gastrointestinal; G3 = grade 3; PE = pulmonary embolism. Data are listed for the specific group when available or the overall cohort if group-specific data are not available.

**Table 3 cancers-15-02359-t003:** Data summary of studies reporting outcomes of patients with spinal and sacral chordomas treated with stereotactic radiation therapy.

Author, Journal, Year Published	Study Type	Number of Patients	Median Follow-Up (mo)	Extent of Resection	Radiation Timing	Prescription Dose (Range)/Number of Fractions	Local Control	Overall Survival	Toxicity
Henderson, Neurosurgery, 2009 [54]	Retrospective	11 (15 targets)	46	Biopsy, STR and/or GTR (margins NR)	RT alone, adjuvant	Median 35 Gy (24–40)/4–5 fractions	5 yr 59.1%	5 yr 74.3%	Hypersthesia with radiculopathy and transient paresthesias in one patient (received 37.5 Gy to cord); abdominal infections in two patients after neoadjuvant SBRT; no other complications attributable to SBRT in patients with spinal or sacral chordoma
Yamada, Neurosurgery, 2013 [55]	Retrospective	24	24	Biopsy, STR	RT alone, neoadjuvant, adjuvant	Median 24 Gy (18–24)/one fraction	Actuarial 95%	Crude 66%	100% G1–G2 odynophagia in patients with cervical or mid thoracic tumors; 13% fracture of lumbar spine or sacrum; 4% sciatic neuropathy (tumor involved sciatic nerve); 4% vocal cord paralysis
Chang, Neurol Res, 2014 [56]	Retrospective	11	50	Biopsy, STR and/or GTR (margins NR)	RT alone, adjuvant	Median 35 Gy (30–50)/3–6 fractions (median 3)	Crude 45%	Mean 84 mo (95% CI: 71–97)	NR
Jung, Technol Cancer Res Treat, 2017 [57]	Retrospective	8 (12 targets)	10	Biopsy, STR and/or GTR (margins NR)	RT alone, adjuvant	Median 16 Gy (11–16)/one fraction	Crude 75%	NR	13% G2 spinal cord myelopathy (resolved with steroids)
Lockney, Neurosurg Focus, 2017 [58]	Retrospective	12	26	Cytoreductive separation surgery	Adjuvant	Median 24 Gy (24–36)/1–3 fractions (median 1)	Upfront (*n* = 5): crude 80% Salvage (*n* = 7): crude 57.1%	All: mean 77.6 mo Upfront: 76.6 mo Salvage: 68.6 mo	27% RT-associated major complications (dysphagia, mucositis, vocal paralysis)
Lu, Rep Pract Oncol Radiother, 2019 [59]	Retrospective	26	44	STR, GTR	Adjuvant	Mean 22.6 Gy/two fractions	5 yr 18.3% (95% CI: 3.0–33.6)	5 yr 59.3% (95% CI: 34.1–84.5)	8% acute G1 skin, GI, and urinary toxicity; 8% acute G2 skin and GI toxicity; no acute G3+ or late G1+ after SBRT
Jin, J Neurosurg Spine, 2020 [60]	Retrospective	35	39	Biopsy, STR, GTR, separation surgery	RT alone, neoadjuvant, adjuvant	Median 24 Gy (18–24)/one fraction	5 yr 80.5% (95% CI: 64.4–96.5)	5 yr 84.3%	31% late G2+; 20% late G3 (tissue necrosis, recurrent laryngeal nerve palsy, myelopathy, fracture, secondary malignancy)
Chen, J Neurosurg Spine, 2021 [61]	Retrospective	28 (30 targets)	21	Biopsy, STR, GTR	RT alone, neoadjuvant, adjuvant	Median 40 Gy (15–50)/1–5 fractions (median 5)	2 yr 96% (95% CI: 74–99)	2 yr 92% (95% CI: 71–98)	12% G3 wound complications in neoadjuvant SBRT arm; 4% G2 PE; 4% G2 stroke; 4% G3 large bowel obstruction; 4% G3 empyema (away from RT field)

Abbreviations: mo = months; STR = subtotal resection; GTR = gross total resection; NR = not reported; RT = radiation therapy; yr = year; SBRT = stereotactic body radiation therapy; G1 = grade 1; G2 = grade 2; CI = confidence interval; GI = gastrointestinal; G3 = grade 3; PE = pulmonary embolism. Data are listed for the specific group when available or the overall cohort if group-specific data are not available.

**Table 4 cancers-15-02359-t004:** Data summary of studies reporting outcomes of patients with spinal and sacral chordomas treated with radiation therapy alone.

Author, Journal, Year Published	Study Type	Number of Patients	Median Follow-Up (mo)	Treatment Intent	Radiation Modality	Prescription Dose (Range)/Dose per Fraction	Local Control	Overall Survival	Toxicity
Chen, Spine, 2013 [8]	Retrospective	24	56	Definitive	Photon, proton	Median 77.4 Gy RBE (70.2–79)/1.8–2.5 Gy RBE fractions	5 yr 79.8%	5 yr 78.1%	33% sacral insufficiency fractions (none requiring surgery); 4% secondary malignancy; 4% erectile dysfunction; 4% perineal numbness; 8% worsening fecal/urinary incontinence; 17% grade 2 rectal bleeding (none requiring new colostomy)
Imai, Br J Radiol, 2011 [47]	Retrospective	84	42	Definitive	Carbon ion	Median 70.4 Gy RBE (52.8–73.6)/3.3–4.6 Gy RBE fractions	5 yr 86%	5 yr 88%	2% skin or soft tissue complications requiring skin graft; 16% severe sciatic nerve complications requiring medication
Imai, IJROBP, 2010 [51]	Phase 1–2 and 2	30	80	Definitive	Carbon ion	Median 70.4 Gy RBE (52.8–73.6)/3.3–4.6 Gy RBE fractions	5 yr 89%	5 yr 86%	5% skin or soft tissue complications requiring skin graft
Yamada, Neurosurgery, 2013 [55]	Retrospective	10	28.5	Definitive, recurrent, metastatic	SBRT	Median 24 Gy (18–24)/18–24 Gy fractions	2 yr 100%	NR	100% G1–G2 odynophagia in patients with cervical or mid thoracic tumors; 13% fracture of lumbar spine or sacrum; 4% sciatic neuropathy (tumor involved sciatic nerve); 4% vocal cord paralysis
Bostel, Radiat Oncol, 2020 [52]	Retrospective	28	60.3	Definitive, recurrent	Carbon ion +/− IMRT	Median 80 Gy RBE (range, 68.8–96 Gy RBE)	5 yr 62%	5 yr 74%	Grade 3 or greater late effects in 21%; Sacral insufficiency fractures in 49% (36% symptomatic); peripheral neuropathy 9%; skin toxicity 9%; intestine 3%
Mima, Br J Radiol, 2014 [28]	Retrospective	23	38	Definitive	Carbon ion or proton	Median 70.4 Gy RBE/2.2 or 4.4 Gy RBE fractions	3 yr 94%	3 yr 83%	Grade 4 dermatitis 22%; grade 3 neuropathy in 17%; grade 3 myositis 9%
Aibe, IJROBP, 2018 [21]	Retrospective	33	37	Definitive	Proton	70.4 Gy RBE/2.2 Gy RBE fractions	3 yr PFS 89.6%	3 yr 92.7%	3% grade 3 acute dermatitis; 3% ileus; 6% pain due to sacral insufficiency fractures
Walser, Clinical Onoclogy 2021 [18]	Retrospective	10	48	Definitive	Proton	Median 74 Gy RBE (range 60–77)/4–4.6 Gy REB fractions	4 yr 77% *	4 yr 85%	7% acute grade 3 dermatitis; 3.5% sacral insufficiency; 1.5% neuropathic pain interfering with ADLs; 3% secondary malignancies
Imai, IJROBP 2016 [46]	Retrospective	188	62	Definitive	Carbon ion	64–73.6 Gy RBE/4–4.6 Gy RBE fractions	5 yr 77.2%	5 yr 81.1%	3% grade 3 toxicity of peripheral nerves; 1% grade 4 skin toxicity
Demizu, Radiat Oncol, 2021 [48]	Retrospective	219	56	Definitive	Carbon ion	67.2 Gy RBE, 70.4 Gy RBE, 79.2 Gy RBE/2.2–4.4 Gy RBE fractions	5 yr 72%	5 yr 84%	1.4% grade 3 myositis; 1% insufficiency fracture; 1% skin disorders; 1% tissue necrosis; 2% grade 4 skin disorders
Evangelisti, Eur Rev Med Pharmacol Sci, 2019 [49]	Prospective	18	23.3	Biopsy	RT alone	70.4 Gy RBE/4.4 Gy RBE fractions	2 yr 84.6%	2 yr 100%	44% late neuropathy; 62.5% grade 1 parasthesia; 37.5% grade 2–3 pain; 5.5% grade 2 late gastrointestinal toxicity

Abbreviations: mo = months; STR = subtotal resection; GTR = gross total resection; NR = not reported; RT = radiation therapy; yr = year; SBRT = stereotactic body radiation therapy; G1 = grade 1; G2 = grade 2; CI = confidence interval; GI = gastrointestinal; G3 = grade 3; PE = pulmonary embolism. Data are listed for the specific group when available or the overall cohort if group-specific data are not available.

**Table 5 cancers-15-02359-t005:** Comparison of preoperative and postoperative radiation therapy for intact and de novo mobile spine/sacral chordomas.

Author, Journal, Year Published	Study Type	Anatomic Location	Number of Patients	Modality	Prescription Dose (Range)/Dose per Fraction	Local Control	Overall Survival
Rotondo, J Neurosurg Spine, 2015 [19]	Retrospective	Mobile spine and sacrum	Preop + Postop: 44 Postop: 51	Proton and photon	Preop 19.8–50.4 GyRBE plus postop to bring dose to 70.2 Gy RBE/1.8–2 Gy RBE fractions	Preop + postop: 5 yr 85%	Pre-op + postop: 5 yr 85%
					Postop: 77.4 GyRBE (range 70.2–77.4 GyRBE)/1.8–2 Gy RBE fractions	Post-op: 5 yr 56%	Post-op: 5 yr 80%
Houdek, J Surg Oncol, 2019 [11]	Retrospective	Sacrum	Preop: 30 Postop: 17 Preop + Postop: 42	Proton and photon	Preop: 50 Gy/1.8–2 Gy RBE fractions Postop: 60.2 +/− 9.9 Gy/1.8–2 Gy RBE fractions Preop + Postop: 70.9 +/− 5.7 Gy RBE/1.8–2 Gy RBE fractions	Not individually reported	Not individually reported
Chen, J Neurosurg Spine, 2021 [61]	Retrospective	Mobile spine and sacrum	Preop: 17 Postop: 5	SBRT	Preop: 40–50 Gy in 5 fractions, 18–21 Gy in 3 fractions, or 16 Gy in 1 fraction	Pre-op 100%	Not individually reported
					Postop: 40 Gy (range 30–50 Gy) in 5 fractions	Postop 80%	
Jin, J Neurosurg Spine, 2019 [60]	Retrospective	Mobile spine and sacrum	Preop: 12 Postop: 11	SBRT	24 Gy (range 18–24 Gy/18–24 Gy fractions	Preop: 3-year LRFS 90% for sacral lesions Individual information not available for postop group	Not individually reported

Abbreviations: mo = months; STR = subtotal resection; GTR = gross total resection; NR = not reported; RT = radiation therapy; yr = year; SBRT = stereotactic body radiation therapy; G1 = grade 1; G2 = grade 2; CI = confidence interval; GI = gastrointestinal; G3 = grade 3; PE = pulmonary embolism. Data are listed for the specific group when available or the overall cohort if group specific data are not available.

**Table 6 cancers-15-02359-t006:** Ongoing and completed but not published clinical trials.

Title and identifier	Sponsor	Phase	Recruitment Status	Estimated Enrollment	Estimated Completion Date	Primary Endpoint	Arms
Nilotinib With Radiation for High Risk Chordoma, NCT01407198	Massachusetts General Hospital	I	Active, not recruiting	29	December 2025	DLTs when treated above the maximum tolerated dose	Nilotinib + EBRT 50.4 Gy
BN Brachyury and Radiation in Chordoma, NCT03595228	Bavarian Nordic	II	Active, not recruiting	29	January 2022	Clinically meaningful objective response rate	BN-Brachyury + radiation
Sacral Chordoma: Surgery Versus Definitive Radiation Therapy in Primary Localized Disease (SACRO), NCT02986516	Italian Sarcoma Group	NA	Recruiting	100	September 2022	Relapse-free survival	Patients who agree to be randomized will receive surgery vs. definitive RT (carbon ion radiotherapy, proton therapy, mixed photons–proton therapy). Those who do not agree to randomization will choose their modality.
Nivolumab With or Without Stereotactic Radiosurgery in Treating Patients With Recurrent, Advanced, or Metastatic Chordoma, NCT02989636	Johns Hopkins University	I	Recruiting	33	March 2022	Incidence of dose limiting toxicities	Arm I: Nivolumab; Arm II: Nivolumab + SRS
Ion Irradiation of Sacrococcygeal Chordoma (ISAC), NCT01811394	Heidelberg University	II	Recruiting	100	June 2022	Safety and feasibility based on incidence of G3 = 5 toxicity	Arm I: 16 × 4 GyE protons; Arm II: 16 × 4 GyE carbon ions
QUILT-3.011 Phase 2 Yeast-Brachyury Vaccine Chordoma, NCT02383498	NantCell, Inc.	II	Active, not recruiting	55	March 2020	Proportion of patients whose tumors shrunk after therapy	Arm I: Radiation (SOC) + GI-6301 Vaccine + Actigraph; Arm II: Radiation + GI-6301 Placebo + Actigraph
Proton Beam Therapy for Chordoma Patients, NCT00496119	MD Anderson Cancer Center	II	Active, not recruiting	15	December 2024	Time to local recurrence	Arm I: 70 GyE PBT at 2 Gy/fx; Arm II: 70 GyE at 2 Gy/fx but using proton beam therapy combined with photon RT where combination improves final dose distribution. Both arms are treated with RT 2+ weeks after surgery.
Improvement of Local Control in Skull Base, Spine and Sacral Chordomas Treated by Surgery and Protontherapy Targeting Hypoxic Cells Revealed by [18F]FAZA) PET/CT Tracers (PROTONCHORDE01), NCT02802969	Institut Curie	II	Recruiting	64	February 2024	Improvement of local control according to RECIST criteria	In residual chordoma after surgery, 78 GyE proton beam therapy (70 GyE to tumor bed and macroscopic volume guided by conventional imaging (CT/MRI) and 8 GyE boost to hypoxic component guided by FAZA (PET/CT)
Hypoxia-positron Emission Tomography (PET) and Intensity Modulated Proton Therapy (IMPT) Dose Painting in Patients With Chordomas, NCT00713037	Massachusetts General Hospital	NA	Completed	20	June 2016	Evaluate if FMISO-PET is a feasible approach for the visualization of hypoxia in skull-base and spinal chordoma	Proton beam therapy + (18F)-FMISO/CT 2 weeks before PBT and 3 weeks after first PBT fraction after 24–36 GyE
Proton Therapy for Chordomas and/or Chondrosarcomas (CH01), NCT00797602	University of Florida	Observational	Completed	189	December 2015	Tumor control	Proton beam therapy
Proton Radiation for Chordomas and Chondrosarcomas, NCT01449149	University of Pennsylvania	NA	Active, not recruiting	50	December 2026	Feasibility	Proton beam therapy 72 to 79.2 Gy RBE in 40–44 fractions
Charged Particle RT for Chordomas and Chondrosarcomas of the Base of Skull or Cervical Spine, NCT00592748	Massachusetts General Hospital	I/II	Completed	381	May 2015	Acute toxicity	Arm I: 40–44 treatments of charged particles; Arm II: 37–40 treatments of charged particles (most will be given with protons but may receive a small portion of photons to spare skin)
Image Assisted Optimization of Proton Radiation Therapy in Chordomas and Chondrosarcomas (CHIPT), NCT04832620	Leiden University Medical Center	Observational	Recruiting	40	November 2023	Determine if functional MRI parameters change within 6 months, and earlier than volumetric changes after start of proton beam therapy, determined by Volumetric and functional MR imaging parameters including permeability parameters	Proton beam therapy + volumetric and functional MR
Randomized Carbon Ions vs. Standard Radiotherapy for Radioresistant Tumors (ETOILE), NCT02838602	Hospices Civils de Lyon	NA	Recruiting	250	December 2026	Progression free survival	Arm I: Carbon; Arm II: photons or proton beam therapy
High Dose Intensity Modulated Proton Radiation Treatment +/− Surgical Resection of Sarcomas of the Spine, Sacrum and Base of Skull, NCT01346124	Massachusetts General Hospital	NA	Active, not recruiting	64	March 2032	Local control	Intensity modulated proton therapy
Comparing Carbon Ion Therapy, Surgery, and Proton Therapy for the Management of Pelvic Sarcomas Involving the Bone, the PROSPER Study, NCT05033288	Mayo Clinic	Observational	Not yet recruiting	180	August 2028	Patient-reported outcome—health-related quality of life; local control	Carbon, protons, surgery (non-randomized)

**Table 7 cancers-15-02359-t007:** Spine Tumor Academy recommendation summary.

Spine Tumor Academy Recommendations	
Recommendation	Level of Strength of Evidence Recommendation
The best chance of cure for mobile spine and sacral chordoma is in the upfront setting. As such, multi-disciplinary expert involvement at time of initial diagnosis is essential to optimizing patient outcomes	III Consensus
Target delineation should be performed on CT scans with at minimum a co-registered T2 weighted MRI. For patients treated in the adjuvant setting the pre-operative T2 weighted MRI should similarly be co-registered. In the adjuvant setting, a comprehensive discussion between the spine surgeon and radiation oncologist should occur to review intraoperative surgical findings and highlight regions believed to be at high risk of recurrence, which may not be obvious based on imaging alone. In the neoadjuvant setting, the discussion should include a review of the surgical plans and intentions to sacrifice or preserve specific nerves in the operating room so that the dosimetric parameters may be adjusted accordingly.	III Consensus
Comprehensive target volumes that include regions of potential microscopic spread have superior local control to focal targets. For SBRT, target delineation according to the consensus contouring guidelines for solid tumor spinal metastases should be considered [63,64]. For proton and heavy ion therapy, comprehensive target delineation is based upon the Massachusetts General Hospital (MGH) Phase 2 data consisting of creation of a low-risk “microscopic” clinical target volume (CTV1) treated to a dose of 19.8–50.4 GyRBE (preoperatively) or 50.4 GyRBE). This is followed by a sequential boost to the high risk CTV2 to 70.2 GyRBE as defined by the original GTV (anatomically constrained) plus 5 mm. A further boost to gross residual disease without margin is performed after maximum safe resection and/or to the definitive GTV to 73.8–77.4 GyRBE [4,19]. PTV is institution specific based upon robustness and range uncertainty analysis.	III Consensus
Although high-level data comparing outcomes comparing dose/fractionation regimens and treatment modalities are unavailable, dose escalation is critical in optimizing local control. Reasonable dose/fractionation schedules by treatment modality include the following: 75.6–77.4 Gy RBE in 1.8–2 Gy RBE fractions using proton +/− photon therapy; 24 Gy in a single fraction or 40–50 Gy in five fractions of SBRT; At least 70.4 Gy in 2.2–4.4 Gy RBE fractions using carbon ion therapy.	III Consensus
When utilizing proton and heavy ion therapy, efforts must be made to limit the dose to the skin to less than 66 GyRBE in order to minimize the risk of long-term wound healing complications [3].	II Consensus

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
