# Peer review of "Radiotherapy for Mobile Spine and Sacral Chordoma: A Critical Review and Practical Guide from the Spine Tumor Academy"

_cancers, 2023, doi:10.3390/cancers15082359_

Round 1

Reviewer 1 Report

The authors write a comprehensive review on the chordoma of Mobile Spine and Sacrum. They used the PRISMA approach and evaluated several aspect of diagnosis and treatment of chordoma. 

The review is very interesting and with fluid style. Tables are exhaustive and well summarize the studies considered. The authors consider treatment with protions, carbon ions and SRT. The overview on surgery is adequate and explains the surgical approach to chordoma. 

They also conclude with recommendations on radiation treatment doses and volumes.   

The treatment of spine and sacrum chordoma is of actual interest and in my opinion this review is important in setting the stage for multidisciplinary and radiotherapy treatment of chordoma. 

The review is suitable for publication in the present form. 

Author Response

We appreciate your kind comments.  We have proofread the document and corrected the English as deemed appropriate. 

Reviewer 2 Report

The authors aimed to summarize the current literature specific to radiotherapy in the management of spine and sacral chordoma, provideing consensus recommendations on behalf of the Spine Tumor Academy.

The paper is well written, the literature search is comprehensive, results are well summarized. 

However, considering that chordoma is a rare tumor and usually managed in centers with high expertise, the scientific sound of the paper is reduced.

Recommendations are provided at the end of the paper. Even though they seem appropriate, there are no details regarding the metodology to achieve such recommendations: in the present form, the paper is more a comprehensive expert review than a formal Consensus Guideline. The authors should improve this part of the paper or rephrase the title.

Despite the provided overview on imaging and surgery could be useful and interesting for non-expert readers, it seems to reduce the ease reading of the manuscript. Moreover, the main topic of the manuscript is radiation therapy. I suggest to reduce or remove these paragraphs.

Minor remarks - Paragraph 5.4 Please, check the typing character

Author Response

We thank you for your careful review and comments.  A point by point response is below:

Comment 1:

The authors aimed to summarize the current literature specific to radiotherapy in the management of spine and sacral chordoma, providing consensus recommendations on behalf of the Spine Tumor Academy. The paper is well written, the literature search is comprehensive, results are well summarized.  However, considering that chordoma is a rare tumor and usually managed in centers with high expertise, the scientific sound of the paper is reduced.

Response 1:

Thank you for your kind comments.  We agree that chordomas represent a relatively rare malignancy.  Nonetheless, given the paucity of data and consensus recommendations regarding this topic, we feel that this manuscript will represent an important and impactful contribution to the literature.

Comment 2:

Recommendations are provided at the end of the paper. Even though they seem appropriate, there are no details regarding the methodology to achieve such recommendations: in the present form, the paper is more a comprehensive expert review than a formal Consensus Guideline. The authors should improve this part of the paper or rephrase the title.

Response 2:

We have revised the methods section to clarify the methodology for providing a new consensus. Specifically, we explain:

Level of agreement regarding the recommendations outlined in the guidelines were defined as follows: 1) consensus: selected by at least 75% of respondents; 2) predominant: selected by at least 50% of respondents; 3) controversial: no single response selected by a majority of respondents. Descriptive statistics were used to review the results.

Comment 3:

Despite the provided overview on imaging and surgery could be useful and interesting for non-expert readers, it seems to reduce the ease reading of the manuscript. Moreover, the main topic of the manuscript is radiation therapy. I suggest to reduce or remove these paragraphs.

Response 3:

We have substantially shortened these sections.  We are happy to eliminate them altogether if that is deemed best.

Comment 4:

Minor remarks - Paragraph 5.4 Please, check the typing character

Response 4:

Thank you for noticing this detail.  The typing character has been corrected.

Reviewer 3 Report

In this comprehensive meta-analysis, the authors evaluated a large number of relevant studies on primary therapeutic approaches in a very rare disease such as chordoma. The authors classified the studies according to therapy-specific approaches such as neoadjuvant vs. adjuvant approach, primary definitive RT vs. adjuvant RT, timing of RT. In addition, the RT techniques such as particle therapy vs. photon RT were analyzed in detail. Furthermore, the dosages and fractionations for each technique were evaluated. 

This topic is very interesting for the radiation oncology community because there is always not much data available to guide treatment. The manuscript is clearly structured and very well written. The tables are well formatted and of high quality. 

In my opinion, the manuscript is suitable for publication.

Author Response

Thank you for your kind comments.

Reviewer 4 Report

In the title, the authors stated this article would provide guidelines on radiotherapy in spinal chordomas.

However, this paper only provides a systematic review mixed with a narrative review. The main issue is the lack of methodology for providing a new consensus (for example, Delphi consensus, expert voting on questions, etc.). Conclusions do not provide recommendations - it's just a straightforward summary of the performed systematic review and opinions/recommendations of other authors (see Table 7).

Moreover, the authors stated that "the primary outcome measure was the rate of local control". Thus, I expected any kind of statistics, meta-analysis, or something similar.

In "radiotherapy guidelines" I hoped for delineation and registration guidelines but this work only focuses on doses and fractionation regimens.

Honestly, I am aware of naming this paper "guidelines" or "consensus".  It is not useful in clinical practice because it does not provide any new data and is too messy to serve as a basis for clinicians.

I kindly suggest that the paper in its present form should not be approved for publication, and I recommend rejection.

Author Response

Thank you for the feedback.  A point by point response is below:

Comment 1:

In the title, the authors stated this article would provide guidelines on radiotherapy in spinal chordomas. However, this paper only provides a systematic review mixed with a narrative review. The main issue is the lack of methodology for providing a new consensus (for example, Delphi consensus, expert voting on questions, etc.). Conclusions do not provide recommendations - it's just a straightforward summary of the performed systematic review and opinions/recommendations of other authors (see Table 7).

Response 1:

We have revised the methods section to clarify the methodology for providing a new consensus. Specifically, we explain:

Level of agreement regarding the recommendations outlined in the guidelines were defined as follows: 1) consensus: selected by at least 75% of respondents; 2) predominant: selected by at least 50% of respondents; 3) controversial: no single response selected by a majority of respondents. Descriptive statistics were used to review the results.

Comment 2:

Moreover, the authors stated that "the primary outcome measure was the rate of local control". Thus, I expected any kind of statistics, meta-analysis, or something similar.

Response 2:

As the intention of this manuscript is to provide consensus guidelines rather than perform a meta-analysis we have removed the statement regarding the primary outcome measure.

Comment 3:

In "radiotherapy guidelines" I hoped for delineation and registration guidelines but this work only focuses on doses and fractionation regimens.

Response 3:

To address this concern, we have added additional recommendations regarding image registration for target delineation:

Target delineation should be performed on CT scans with at minimum a co-registered T2 weighted MRI.  For patients treated in the adjuvant setting the pre-operative T2 weighted MRI should similarly be co-registered.  In the adjuvant setting, a comprehensive discussion between the spine surgeon and radiation oncologist should occur to review intraoperative surgical findings and highlight regions believed to be at high risk of recurrence which may not be obvious based on imaging alone. In the neoadjuvant setting, the discussion should include a review of the surgical plans and intentions to sacrifice or preserve specific nerves in the operating room so that the dosimetric parameters may be adjusted accordingly.

In addition, we include recommendations regarding target volumes as follows:

Comprehensive target volumes that include regions of potential microscopic spread have superior local control to focal targets. For SBRT, target delineation according to the consensus contouring guidelines for solid tumor spinal metastases should be considered  [67,68].  For proton and heavy ion therapy, comprehensive target delineation is based upon the Massachusetts General Hospital (MGH) Phase 2 data consisting of creation of a low risk “microscopic” clinical target volume (CTV1) treated to a dose of 19.8 – 50.4 GyRBE (preoperatively) or 50.4 GyRBE). This is followed by a sequential boost to the high risk CTV2 to 70.2 GyRBE as defined by the original GTV (anatomically constrained) plus 5mm. A further boost to gross residual disease without margin is performed after maximum safe resection and/or to the definitive GTV to 73.8 – 77.4 GyRBE [8,23]. PTV is institution specific based upon robustness and range uncertainty analysis.

Comment 4:

Honestly, I am aware of naming this paper "guidelines" or "consensus".  It is not useful in clinical practice because it does not provide any new data and is too messy to serve as a basis for clinicians. I kindly suggest that the paper in its present form should not be approved for publication, and I recommend rejection

Response 4:

We have made extensive revisions to the manuscript to incorporate your excellent feedback and suggestions.  We are hopeful that in its revised form the manuscript will be suitable for publication in your prestigious journal.

Round 2

Reviewer 2 Report

The authors did address all the raised issue: the ease reading of the manuscript was improved; methods regarding the reported recommendations are clear in the present form.

Minor comment: Section 2.2, line 115, "were were", please check

Author Response

Comment:

Minor comment: Section 2.2, line 115, "were were", please check

Response:

Thank you for identifying this typo.  It has been corrected. 

Reviewer 4 Report

Dear Authors,

The manuscript has slightly improved but it's still not a consensus or guidelines.

You added the methodology of expert voting but you did not provide any results. It looks a bit artificial.

Moreover, as another reviewer suggested,

This manuscript, in my opinion (the Editor and Authors might have a different view of course), is overtalked and not concise. The parts regarding imaging, surgery, etc. are unnecessary. 

Take a look at other consensuses:

https://www.estro.org/Science/Guidelines

https://www.practicalradonc.org/cms/10.1016/j.prro.2021.04.005/attachment/87216780-c88f-4be4-9a20-d1e3e4cf643b/mmc1.pdf

Author Response

Comment 1:

The manuscript has slightly improved but it's still not a consensus or guidelines.

Response 1:

Thank you for your feedback regarding our manuscript.  Revising the manuscript into the PICO format would require restarting from the beginning and ultimately be a very different manuscript from the one that we have written.  We feel that this would be an excellent future project for the Spine Tumor Academy.  Instead, in response to your concern we have revised the title and verbiage throughout the text to call it “A Critical Review and Practical Guide From the Spine Tumor Academy” rather than "Radiotherapy Consensus Guidelines".

Comment 2:

You added the methodology of expert voting but you did not provide any results. It looks a bit artificial.

Response 2:

We have added a “Strength of Recommendation” column to Table 7 to present the results of the voting process.

Of note, we feel that the voting method has strengthened the manuscript even in its new format as a review article rather than consensus guidelines.  However, we are willing to remove these methods and results if deemed best by the reviewers and/or editors.

Comment 3:

Moreover, as another reviewer suggested,

This manuscript, in my opinion (the Editor and Authors might have a different view of course), is overtalked and not concise. The parts regarding imaging, surgery, etc. are unnecessary. 

Response 3:

We have removed the imaging and surgical sections from the manuscript as requested.

Round 3

Reviewer 4 Report

Finally, after the implementation of the reviewers' suggestions the manuscript has improved - thank you.

The only thing you may have missed during your search is the lack of some important articles regarding RT in chordomas - please consider including and discussing other comprehensive reviews, like https://doi.org/10.1016/j.ejso.2020.04.028

Author Response

Comment:

The only thing you may have missed during your search is the lack of some important articles regarding RT in chordomas - please consider including and discussing other comprehensive reviews, like https://doi.org/10.1016/j.ejso.2020.04.028

Response:

The literature search for this manuscript intentionally included primary research citations only and excluded review articles.  However, at your request and for the ease of the reader, we have added references to the following previous reviews which have been published in the last decade:

  1. Radaelli S.; Fossati P.; Stacchiotti S.; Akiyama T.; Ascencio J.M.; Bandiera S.; Boglione A.; Boland P.; Bolle S.; Bruland O. et al. The Sacral Chordoma Margin. Euro J. Surg. Onc. 2020,46(8),1415-1422.
  2. Pennington Z.; Ehresman J.; McCarthy E.F.; Ahmed A.K.; Pittman P.D.; Lubelski D.; Goodwin C.R.; Sciubba D.M. Chordoma of the Sacrum and Mobile Spine: A Narrative Review. Spine J. 2021,21(3),500-517.
  3. Ailon T.; Torabi R.; Fisher C.G.; Rhines L.D.; Clarke M.J.; Bettegowda C.; Boriani S.; Yamada Y.J.; Kawahara N.; Varga PP, et al. Management of Locally Recurrent Chordoma of the Mobile Spine and Sacrum: A Systematic Review. Spine. 2016,41,S193-198.
  4. Walcott B.P.; Nahed B.V.; Mohyeldin A.; Coumans J.V.; Kahle K.T.; Ferreira M.J. Chordoma: Current Concepts, Management, and Future Directions. Lancet Oncol. 2012,13(2),e69-76.